# Enhanced Cataluminescence Sensor Based on SiO_2_/MIL-53(Al) for Detecting Isobutylaldehyde

**DOI:** 10.3390/molecules29143287

**Published:** 2024-07-11

**Authors:** Qianchun Zhang, Xixi Long, Shan Tang, Li Jiang, Zhaoru Ban, Yanju Chen, Runkun Zhang

**Affiliations:** 1Key Laboratory for Analytical Science of Food and Environment Pollution of Qian Xi Nan, School of Biology and Chemistry, Xingyi Normal University for Nationalities, Xingyi 562400, China; longxixi@xynun.edu.cn (X.L.); tangshan@xynun.edu.cn (S.T.); jiangli@xynun.edu.cn (L.J.); banzhaoru@xynun.edu.cn (Z.B.); chenyanju@xynun.edu.cn (Y.C.); 2Guangdong Provincial Engineering Research Center of Public Health Detection and Assessment, School of Public Health, Guangdong Pharmaceutical University, Guangzhou 510310, China

**Keywords:** cataluminescence, SiO_2_/MIL-53(Al), catalysis, isobutylaldehyde sensor, GC–MS, density functional theory

## Abstract

A simple, rapid, and reliable method for detecting harmful gases is urgently required in environmental security fields. In this study, a highly effective cataluminescence sensor based on SiO_2_/MIL-53(Al) composites was developed to detect trace isobutylaldehyde. The sensor was designed using isobutylaldehyde to generate an interesting cataluminescence phenomenon in SiO_2_/MIL-53(Al). Under optimized conditions, a positive linear relationship was observed between the signal intensity of the cataluminescence and isobutylaldehyde concentration. The isobutylaldehyde concentration range of 1.55–310 ppm responded well to the sensing test, with an excellent correlation coefficient of 0.9996. The minimum detectable concentration signal-to-noise ratio (*S/N* = 3) was found to be 0.49 ppm. In addition, the sensor was effectively utilized for analyzing trace isobutylaldehyde; the analysis resulted in recoveries ranging from 83.4% to 105%, with relative standard deviations (*RSDs*) of 4.8% to 9.4%. Furthermore, the mechanism of cataluminescence between SiO_2_/MIL-53(Al) and isobutylaldehyde was explored using GC–MS analysis and density functional theory. We expect that this cataluminescence methodology will provide an approach for the environmental monitoring of isobutylaldehyde.

## 1. Introduction

Isobutylaldehyde (IBL) is an essential volatile organic compound (VOC) that has been extensively used as a raw material in the organic chemical industry. It can be used to synthesize a variety of high-value chemical products, such as isobutyric acid, isobutanol, acetals, oximes, and imines [1]. IBL gas is risky because it is combustible and reacts violently with oxides. It also exhibits a high spreading rate, which increases the risk of rapid spread. Consequently, it is crucial to prioritize the advancement of IBL detection techniques that are sensitive, dependable, and swift, with regard to both safeguarding the environment and ensuring human well-being.

Although traditional GC, GC–MS [2], and high-performance liquid chromatography with complex derivatization [3] are highly sensitive methods for detecting trace amounts of IBL, they are often expensive, cumbersome, and time-consuming [4,5]. Cataluminescence (CTL) is a special chemiluminescence emitted by compounds when they are catalyzed on the surface of a solid catalyst. In 1976, Breysse and others first discovered CTL [6], bringing a new insight to the scientific community. Subsequently, CTL sensors have been extensively used and developed. For the past few years, there has been rapid progress in the development of CTL for the detection of trace IBL. Our research group has undertaken preliminary studies on IBL detection based on CTL [7]; CTL-based sensing detection is a long-term, effective, and easy-to-use method for monitoring harmful gases [8,9,10,11,12]. CTL sensors may generate considerable interest in the domain of gas sensing, owing to their advantages over the traditional chemiluminescence systems based on irreversible reactions, low cost, simplicity [13,14,15], rapid response, excellent selectivity, and high sensitivity [11,16]. CTL has the potential for gas sensing and VOC detection application. As remarkable developments have been obtained that apply to oxides such as ZrO_2_:Eu^3+^ [17], MnO_2_ [18], CeO_2_ [19], CuO [20], and SiO_2_ [21] as sensing materials in CTL, a considerable effort has been devoted to exploitative novel sensing materials that exhibit outstanding CTL performance, which helps further the selectivity and sensitivity of sensors to a higher degree. Recently, composites with excellent CTL performance have been widely studied [22,23,24,25,26,27,28,29]. The composite material may increase the overall porosity and surface area, which may offer additional sites for interaction with analyte gases [30]. SiO_2_, which is one of the most abundant substances on Earth, is widely used as catalyst support. Choi and his team’s research results show that the mesoporous SiO_2_/Au films can be applied to detect acetone, formaldehyde, and methane gas [31]. Additionally, Ge et al. and Joo et al. [32,33] demonstrated increased gas sensing capabilities through the use of α-Fe_2_O_3_/SiO_2_ nanocomposites and In_2_O_3_-SiC-SiO_2_ composites, respectively. Metal–organic frameworks (MOFs) are hybrid materials synthesized via the assembly of metal cations or through the loss of two univalent or a bivalent atoms or group and organic connections. They exhibit high crystallinity and advantageous favorable properties with extensive porosity and a distinctive network structure; these features make them highly versatile and suitable for a broad range of applications [34,35] such as gas adsorption and separation [36,37], catalysis [38,39], and chemical sensing [40,41]. In particular, they have been applied to ameliorate the performance of CTL sensors [42]. Zou et al. were the first to use MOFs as catalytic materials for CO-catalytic oxidation reactions. However, high temperatures tend to decompose MOFs and destroy their structures. The low thermal stability of MOFs limits their use in catalysis. Therefore, improving the catalytic performance remains an exciting challenge for achieving high stability and outstanding catalytic performance. Serre et al. [43] synthesized MIL-53(Al), a well-studied MOF, which exhibited excellent catalytic performance. Once SiO_2_ is introduced as a dopant, SiO_2_/MIL-53(Al) transforms into a highly effective sensing material that not only showcases exceptional catalytic capabilities but also demonstrates remarkable stability when contrasted with standalone single-component substances. These attributes position them as ideal candidates for sensors.

A strong CTL response was shown when the IBL was obtained through the surface of SiO_2_/MIL-53(Al). Based on this particular phenomenon, an IBL sensor was developed utilizing a SiO_2_/MIL-53(Al) CTL, which exhibited excellent performance. Furthermore, the CTL sensing of IBL on SiO_2_/MIL-53(Al) was further researched through employing density functional theory (DFT) and GC–MS analyses, which provided valuable insights into the fundamental principles of chemiluminescence procedures and catalytic oxidation reactions based on CTL function materials.

## 2. Results and Discussion

### 2.1. Material Characterization

The phases and crystal structures of the materials were analyzed using X-ray diffraction (Rigaku diffractometer, Shimadzu 6100, Kyoto, Japan, XRD) and Fourier transform infrared spectroscopy (Nicolet IS10 spectrometer, Thermo Fisher Scientific, Waltham, MA, USA, FT–IR). The crystal structures of SiO_2_, MIL-53(Al), and SiO_2_/MIL-53(Al) were examined using XRD. Both MIL-53(Al) and SiO_2_/MIL-53(Al) exhibit identical diffraction peaks at 18.03°, 23.50°, 25.21°, 27.41°, 33.80°, 37.28°, and 44.25°, consistent with those reported in [44,45]. The XRD patterns of SiO_2_ and MIL-53(Al) revealed their crystal structures (Figure 1A(a,b)). Figure 1A(c) illustrates that the XRD patterns of SiO_2_/MIL-53(Al) crystal planes, including (002), (200), (−202), (−103), (301), (020), and (303), show a resemblance to MIL-53(Al), suggesting that the crystal plane of SiO_2_/MIL-53(Al) remains largely unchanged despite the addition of SiO_2_. The FT-IR spectroscopy was utilized to analysed the surface functional groups of the materials in Figure 1B. The SiO_2_/MIL-53(Al) FT-IR spectrum demonstrated analogous characteristic peaks to those of MIL-53(Al) and SiO_2_, which include, due to the presence of Si−OH groups adsorbed on the surface, a peak at 3435 cm^−1^ leading to the unique absorption of oh groups, −C=O asymmetric stretching at 1608 cm^−1^, −COO symmetric stretching vibration at 1418 cm^−1^, an asymmetric stretching vibration of 1117 cm^−1^ belonging to void oxygen in bulk silicon, and a symmetric stretching vibration absorption peak of a Si−O−Si bond of 806 cm^−1^, respectively [46,47].

As is commonly understood, the effectiveness of a catalyst is directly influenced by its surface area. Therefore, to gain deeper insights into the structural properties of the SiO_2_/MIL-53(Al) material, the Brunauer–Emmett–Teller (ASAP 2460 instrument, Boston, MA, USA, BET) analysis was also performed. The specific surface area (SBET), average diameter, and pore volume are shown in Appendix A. The specific surface area of MIL-53(Al) is 304.03 m^2^/g; nevertheless, SiO_2_/MIL-53(Al) exhibited a higher BET surface area (138.92 m^2^/g) than that of SiO_2_ (3.81 m^2^/g), and the good surface area of SiO_2_/MIL-53(Al) provided an appropriate explanation for its excellent CTL activity in IBL-sensing. The isotherms displayed in Figure 1C illustrate that all the SiO_2_/MIL-53(Al) structures exhibit type IV hysteresis loops across a range of *P/P*_0_ values from 0.0098 to 0.9860. Moreover, as depicted in Figure 1D, the average pore sizes for the three samples were determined to be 9.0 nm using the Barrett–Joyner–Halenda calculation method on the adsorption branches. Additionally, the results indicate that the pore volume of SiO_2_/MIL-53(Al) is greater than that of SiO_2_ and MIL-53(Al), with values of 0.306555, 0.008775, and 1.604335 cm^3^/g, respectively. The unique structure of SiO_2_/MIL-53(Al) provided favorable conditions for facilitative gas diffusion on the material surface, thereby improving its sensing performance.

The characterization of the microstructures and morphologies of catalytic materials was carried out by performing Scanning Electron Microscopy (SEM) using the SU8020 spectrometer from Tokyo, Japan, and Transmission Electron Microscopy (TEM) using the FEI Talos F200X from Tokyo, Japan. The SEM images depicted in Figure 2A clearly exhibit that the SiO_2_ particles are spherical in shape, varying in size from around 0.330 to 2.00 µm. The morphology of MIL-53(Al) is also demonstrated using SEM in Figure 2B, which shows an exceedingly interesting hard agglomeration, with the particle diameter being in the range of 130 to 654 nm. The square block structure of MIL-53(Al) can be clearly seen, and a few small particles are observed on the MIL-53(Al) materials, probably due to insufficient washing with DMF in the preparation process. This kind of block structure exhibits a large specific surface area; therefore, oxygen molecules and the test gas molecules are more effectively adsorbed on the catalytic material on the internal, resulting in a rapid response to the target gas diffusion. The SEM images of SiO_2_/MIL-53(Al) are displayed in Figure 2C; it can be seen that the smaller MIL-53(Al) particles adhere to the larger SiO_2_ particles, which better fills the vacancy and creates more contact between the gas and the catalyst, thus improving its catalytic performance. As shown in Figure 2D, the shape of SiO_2_ is uniform and orderly agglomerated, and the particle size of SiO_2_ is approximately 97 to 131 nm. The TEM images of MIL-53(Al) are shown in Figure 2E, which show cubic long strips consisting of uniformly spaced sheets, and the particle size of MIL-53(Al) is approximately 260–500 nm. According to Figure 2F, the result shows that MIL-53(Al) agglomerates around SiO_2_, indicating that SiO_2_ doping does not affect the microstructure of MIL-53(Al).

The stability of catalytic materials is crucial in evaluating their performance, especially when it comes to the possibility of catalytic thermal runaway at elevated temperatures. In this study, we analyzed the thermal stabilities of catalytic materials through the utilization of thermogravimetry (TG). The analysis of the results, as depicted in Appendix A, reveals that within the temperature range of 80–600 °C, SiO_2_ demonstrated no alterations in mass. In contrast, the mass of MIL-53(Al) gradually decreased until it reached a temperature of 168.7 °C, with the most significant loss occurring at 241.9 °C. The TG analysis of SiO_2_/MIL-53(Al) indicated that at the experimental temperature of 177 °C, there was no effect on its mass. The mass starts to reduce at 190 °C, with a greater mass loss at 586.7 °C. As the temperature continues to rise, the SiO_2_/MIL-53(Al) structure was destroyed. Hence, it can be inferred that the thermal stability of SiO_2_ is higher after doping MIL-53(Al) at a certain temperature.

### 2.2. Selection of the Sensing Material and Gases

The sensing material played a crucial role in the response to the monitored substances in the CTL-based sensor system. Figure 3A illustrates the catalytic materials’ performances, namely MIL-53(Al), SiO_2_, and SiO_2_/MIL-53(Al) composites with mass ratios of 1:2, 1:1, and 2:1, respectively. It is evident that the signal value obtained from the SiO_2_/MIL-53(Al) composites is significantly higher than that of MIL-53(Al) and SiO_2_ alone. The CTL intensities of the IBL composites with three different proportions were tested, and the mass ratio of 1:1 exhibited the best catalytic effect. Therefore, the mass ratio of 1:1 was selected as the optimal ratio for the composite material to be used in subsequent analyses. The possible reasons are as follows: (1) Synergistic effects: The interaction between SiO_2_ and MIL-53(Al) can lead to synergistic effects that enhance the overall catalytic activity of the material. (2) Improved stability: The presence of SiO_2_ can potentially improve the stability of MIL-53(Al) under reaction conditions, leading to better catalytic performance over time. (3) Enhanced surface area: The combination of SiO_2_ and MIL-53(Al) can lead to an increased surface area which provides more active sites for catalytic reactions to occur. These factors combined can contribute to the superior catalytic effect of SiO_2_/MIL-53(Al) compared to pure MIL-53(Al) or SiO_2_ alone.

To analyze the catalytic performance of the materials, we compared their performance with those of common catalytic materials. Nano-ZrO_2_, nano-NiO, nano-In_2_O_3_, nano-Sn_2_O_3_, and nano-CuO were used as the sensing materials to study the catalytic performance under the same experimental conditions. A total of 1.0 mL IBL (155 ppm) was injected into the different sensing materials. It was found that SiO_2_/MIL-53(Al) showed the highest CTL activity; the IBL intensities on the surfaces of SiO_2_, MIL-53(Al), and nano-ZrO_2_ were 54.9%, 15.4%, and 3.6% lower than that of SiO_2_/MIL-53(Al), respectively. The smaller CTL responses, or their absence, were observed with other catalysts. These findings highlight the effective catalytic performance of SiO_2_/MIL-53(Al) materials. Consequently, SiO_2_/MIL-53(Al) was selected as the sensing material to further explore the variability of CTL performance with various VOCs, as illustrated in Figure 3B and Appendix A. Twenty-six common VOCs, including aldehydes (IBL, methylallyl aldehyde, methanal, glutaraldehyde), alcohol (propanol, methanol, 3-methyl-2-butene-1-ol, sec-butyl alcohol), ketone (acetone, acetylacetone, methyl isobutyl ketone, cycloheptanone), ether (ethylene glycol butyl ether, diethyl ether, 2-butoxy ethanol), benzene (paraxylene), alkane (hexane, n-pentane), amine (diethylamine, formamide, cholamine), alkene (styrene, chloropropene, tetra-chloroethylene), and ester (ethyl acetate, methyl acetate) were measured using the SiO_2_/MIL-53(Al)-based sensor. It was shown that IBL exhibited the highest CTL intensity. The signal consequence from diethyl ether detection was only 5.30% of that of IBL. Ethyl acetate and propanol also produced weak CTL intensities at only 0.98% and 0.46% of that of IBL, respectively. The other VOCs either produced even weaker or no CTL signal. Therefore, the selective detection of IBL could easily be achieved, demonstrating that the SiO_2_/MIL-53(Al)-based sensor is selective, sensitive, and effective. Thus, SiO_2_/MIL-53(Al) can be used as a catalyst for the detection of IBL.

### 2.3. Optimization of CTL Conditions

Generally, the catalytic oxidation of IBL on the SiO_2_/MIL-53(Al) surface is influenced by temperature, wavelength, and flow rate. To gain the optimal working parameters for the detection of IBL, the following steps were undertaken. Since the temperature would play a vital part on the capability of CTL sensing material, the working temperature of the CTL emission was first evaluated in the range of 136 to 220 °C (136, 157, 177, 199, and 220 °C), via 80 mL min^−1^ air and through a 460 nm filter. As shown in Appendix A, it showed that the CTL intensity or *S/N* arrived to a maximum at 177 °C. A moderate temperature could be conducive to the catalytic oxidation of IBL. However, it is observed that a significant decrease in the *S/N* occurs when the operating temperature surpasses 177 °C. This phenomenon may be attributed to the reduction in thermal radiation as the temperature reaches its peak. Therefore, 177 °C was considered as the first-rank detection temperature for the subsequent research. The CTL intensity of SiO_2_/MIL-53(Al) towards wavelength was measured; as shown in Appendix A, six wavelengths were detected in the range of 412–505 nm (412, 440, 460, 475, 490, and 505 nm) under the optimized working temperature of 177 °C and flow rate of 80 mL min^−1^. The results show that the CTL intensity and *S/N* ratio slightly increased at the beginning, and then declined when the wavelength increased; the superexcellence wavelength was 460 nm. Subsequently, the influence of air flow on CTL intensity was further studied, while air flow rate increased from 60 to 100 mL min^−1^ under the optimized working temperature of 177 °C and the wavelength of 460 nm. As exhibited in Appendix A, the CTL intensity had an obvious increase. Nevertheless, the CTL intensity declined when the flow rates exceeded 80 mL min^−1^; this was because the lower flow rate allows oxidation processes to occur under diffusion-controlled conditions. In conclusion, 80 mL min^−1^ was chosen for the subsequent CTL experiments.

### 2.4. Method of Analysis and Application

Method of analysis: Under optimal experimental conditions (177 °C, detection wavelength = 460 nm, and carrier gas flow rate = 80 mL min^−1^), the analytical characteristics of the CTL IBL sensor based on SiO_2_/MIL-53(Al) were investigated, and the fitted equation was appraised at different IBL concentrations on the surface of SiO_2_/MIL-53(Al). The release response signals of CTL at seven different concentrations of IBL (310, 155, 77.6, 31.0, 15.5, 3.10, and 1.55 ppm) were studied using the gradient dilution of IBL. The results are presented in Figure 4A. The CTL intensity of IBL on the surface of SiO_2_/MIL-53(Al) increased with increasing IBL concentration. Response and recovery times are key parameters for the rapid analysis of gas sensors; the CTL response curve variations over time for different IBL concentrations resembled each other, with the response/recovery times at approximately 0.5 s/5 s. The fitted equation of CTL intensions versus IBL concentrations are shown in Figure 4B. In the concentration range of 1.55–310 ppm, a strong linear correlation was found between the responses and concentrations of IBL. The linear regression equation, represented as *I* = 114.5C − 10.3, exhibited excellent linearity with a coefficient of determination (*R*^2^) of 0.9996. Furthermore, the limit of detection (*S/N* = 3) was determined to be 0.49 ppm. Compared with the previously reported CTL sensor for IBL [7], the one reported herein exhibited higher sensitivity and faster response time. Figure 4C demonstrates that the sensor exhibited a promising level of repeatability, as evidenced by the low relative standard deviation (*RSD*) of 5.6% from ten parallel tests conducted at a consistent concentration of 3.10 ppm. The durability of the sensor that relies on the sensing material was evaluated by subjecting SiO_2_/MIL-53(Al) to a continuous flow of 3.10 ppm IBL at a temperature of 177 °C for a duration of 7 days. As depicted in Figure 4D, the *RSD* (*n* = 7) was found to be 6.4%, indicating that the CTL intensity remained consistently stable over 7 days. Moreover, only one CTL sensor based on SiO_2_/MIL-53(Al) has been utilized to finalize this study. It has shown the capability of being reused over 500 times without any notable alterations in the *S/N*, demonstrating high stability.

Analysis of samples: To assess the practical applicability of the newly developed sensor to IBL, five spiked samples and two real gas samples were used. Five spiked samples with different concentrations of IBL and other VOCs, including cyclohexanone, butanal, isobutanol, ethyl acetate, n-propanal, diethyl ether, methylallyl aldehyde, IBL, 2-butoxy ethanol, styrene, methyl isobutyl ketone, and 3-methyl-2-butene-1-ol were added. Seven 1 L sampling bags were used to collect the five spiked samples, as well as the volatile gas samples in a refrigerated display cabinet and a medicine locker display cabinet. These samples were then measured using the designed sensor, with a total of seven measurements taken. The outcomes were delineated in Table 1, demonstrating recoveries spanning from 83.4% to 105%, accompanied by relative standard deviations (*RSDs*) ranging from 4.8% to 9.4%, which are deemed satisfactory.

### 2.5. Sensing Mechanism

The developed IBL sensor exhibits a remarkable advantage in its superior CTL selectivity towards IBL. This can be attributed to the diverse adsorption energies of various gases onto SiO_2_/MIL-53(Al). To further prove this hypothesis, we derived DFT-computed adsorption configurations and energies for the SiO_2_/MIL-53(Al) sensing mechanism. The adsorption energies of H_3_CCH(CH_3_)CHO, H_2_O, and O_2_ molecules on SiO_2_ (100) were calculated using DFT. The optimized adsorption structures are shown in Figure 5a–c. The adsorption of H_3_CCH(CH_3_)CHO, H_2_O, and O_2_ on the surface of the SiO_2_ (100) was −2.63 and −2.07 and −2.38 eV, respectively. This indicates that the H_3_CCH(CH_3_)CHO, H_2_O, and O_2_ molecules were chemically adsorbed on the SiO_2_ (100) surface. To better understand the change in charges in the adsorption configuration, the differential charge densities of the H_3_CCH(CH_3_)CHO, H_2_O, and O_2_ molecules on the surface of SiO_2_ (100) were analyzed (Figure 5d–f). In the differential charge diagram of H_3_CCH(CH_3_)CHO (Figure 5d), the charge density between the carbon and oxygen atoms of the aldehyde group in the H_3_CCH(CH_3_)CHO molecule decreased. In Figure 5e, the differential charge diagram of H_2_O showed a decrease in charge density on the surface of the H atom near the O atom. Conversely, the charge density on the surface of the H atom closer to the SiO_2_ (100) surface increased. In Figure 5f, the differential charge diagram of the O_2_ molecule indicates a decrease in the area between the two O atoms. In conclusion, the C=O bond in H_3_CCH(CH_3_)CHO, O-H bond in H_2_O, and O-O bond in O_2_ molecules are more likely to be broken in the adsorbed state. Therefore, by combining the results of the GC–MS analysis of the products, we propose a possible mechanism for H_3_CCH(CH_3_)CHO sensing. In addition, the CTL reaction products of IBL on the SiO_2_/MIL-53(Al) surface were investigated using GC–MS. These products can be observed in Appendix A, with seven peaks corresponding to HCHO, CH_3_CHO, CH_3_CH_2_CHO, CH_2_=C(CH_3_)CHO, H_3_CCH(CH_3_)CHO, (CH_3_)_2_CHCOOH, and CH_3_COCH_3_. Based on the detected products, a probable mechanism for CTL in IBL oxidation is devised. When the IBL gas touched the surface of SiO_2_/MIL-53(Al), it was catalyzed by oxygen in the air to form electron-excited intermediates. Subse-quently, the excited intermediates transform into HCHO, CH_3_CHO, CH_3_CH_2_CHO, CH_2_=C(CH_3_)CHO, H_3_CCH(CH_3_)CHO, (CH_3_)_2_CHCOOH, and CH_3_COCH_3_, and emit luminescence.

## 3. Materials and Methods

### 3.1. Chemical Reagents and Materials

The reagents mentioned below were obtained from Aladdin Chemistry Co., Ltd. based in Shanghai, China: Al(NO_3_)_3_·9(H_2_O), Na_2_SiO_3_, acetonitrile, terephthalic acid (H_2_BDC), IBL, propanol, acetone, ethylene glycol butyl ether, paraxylene, hexane, diethylamine, formaldehyde, acetaldehyde, acrolein, isobutyric acid, diethyl ether, ethyl acetate, methyl acetate, n-pentane, acetylacetone, cyclohexanone, butanal, isobutanol, ethyl formate, methylallyl aldehyde, n-propanal, methanal, glutaraldehyde, 3-methyl-2-butene-1-ol, sec-butyl alcohol, methyl isobutyl ketone, cycloheptanone, 2-butoxy ethanol, formamide, cholamine, styrene, chloropropene, tetrachloroethylene, ZrO_2_, NiO, In_2_O_3_, Sn_2_O_3_, and CuO. *N, N*-dimethylformamide (DMF) was obtained from J&K Scientific, Inc. (Beijing, China). All chemical reagents used in the experiments were analytically pure and used as received, with no further purification.

### 3.2. Preparation of SiO_2_/MIL-53(Al) Composites

MIL-53(Al) was synthesized using an extremely simple hydrothermal reaction as follows [48]: 3.75 g (0.0100 mol) of Al (NO_3_)_3_ 9(H_2_O) and 1.66 g (0.0100 mol) of H_2_BDC were added to 65.0 mL of DMF and dispersed using ultrasound for 30 min. The resulting suspension was then removed to a 100 mL Teflon-lined stainless autoclave and kept at 220 °C for 96 h. Once the reaction had finished, the mixture was left to cool to a temperature of 25 ± 5 °C, at which point a white precipitate formed. The precipitate underwent three rounds of washing, using 40 mL of both high purity water and DMF. The resulting compounds were centrifuged at 9500 rpm for 10 min, and subsequently the prepared MIL-53(Al) was dried under vacuum at 80 °C for 24 h. Finally, it was ground into a fine powder.

A simple wet chemical method was used to synthesize the SiO_2_ nanoparticles [49]. First, a 0.1 mol/L solution of HCl was gradually added to a solution of 12.2 g Na_2_SiO_3_, which was maintained at 50 °C in a water bath and continuously stirred until it dissolved completely. After filtering the white precipitate, it was rinsed with ultrapure water seven times. The precursors were subsequently dried in an oven at 80 °C for a duration of 12 h. The product was then subjected to calcination at 300 °C for a period of 1 h. The final materials were obtained by subjecting the product to calcination under 600 °C for 4 h. SiO_2_/MIL-53(Al) was prepared via ultrasonication as follows: MIL-53(Al) (50.0 mg) and SiO_2_ (50.0 mg) were added to 4 mL of anhydrous ethanol and ultrasonicated for 30 min to achieve suspensions of MIL-53(Al) and SiO_2_, respectively. Subsequently, the SiO_2_ suspension was promptly added to the MIL-53(Al) dispersion and ultrasonicated for 30 min to achieve a uniform dispersion. The sensor material was then obtained through filtration and dried for 3 h under vacuum at 80 °C. Following this, 150 µL of ultrapure water was added, and the resulting products were thoroughly stirred to coat a cylindrical ceramic rod. So as to ensure the reproducibility of the obtained SiO_2_/MIL-53(Al), the above was repeated three times.

### 3.3. Instrumentation and Analysis

The BPCL Ultraweak chemiluminescence Analyser from the Biophysics Institute of the Chinese Academy of Science in Beijing, China was used to measure the CTL signal. Appendix A shows a diagrammatical of the CTL gas sensor. A proper detection wavelength and carrier gas flow rate were chosen, with air as the carrier gas. The injection volume was 1.00 mL, which was continuously and stably carried into the CTL reaction cell using an air peristaltic pump. When SiO_2_/MIL-53(Al) activated the catalytic oxidation reactions, the CTL emission was measured using a photomultiplier tube (0.5 s per spectrum was set as the data integration time and 1000 V served as the working voltage of photomultiplier tube). The analysis of the tail gas was conducted utilizing a Trace 1300-ISQ7000 GC–MS system from Thermos Fisher Scientific, Shelton, CT, USA, equipped with an HP-INNO Wax column measuring 30 m in length, 0.32 mm in internal diameter, and a film thickness of 0.25 µm.

### 3.4. Computation of Sensing Mechanism

The Vienna ab initio simulation package (VASP) was utilized for all DFT calculations [50,51]. The Perdew–Burke–Ernzerhof functional was utilized to characterize the core electrons, incorporating the commutative dependence of the generalized gradient approximation. The computation further employed projector-augmented wave pseudopotentials [52]. The energy cut-off value for the plane wave was established as 400 eV. To ensure convergence during self-consistent field iteration, energy and force thresholds were set at 10^−5^ and 0.02 eV/Å, respectively. To avoid interactions between periodic layers, a vacuum of 15 Å was upheld in the z-direction. The sampling of SiO_2_ cells within the Brebruin region was refined through the utilization of a 15 × 15 × 9 Monkhorst–Pack grid, whereas the surface structure calculations utilized a 1 × 1 × 1 Monkhorst–Pack grid. A SiO_2_ (100) [53,54] surface model (containing 72 O atoms and 36 Si atoms) was optimized based on a SiO_2_ cell.

The adsorption energy ($E_{ads}$) is calculated using the following equation:(1)$$E_{ads}=E_{total}-E_{substrate}-E_{molecule}$$
where $E_{total}$ is the total energy of a molecule (H_3_CCH(CH_3_)CHO, H_2_O, or O_2_) adsorbed on the SiO_2_ (100) substrate and $E_{substrate}$ and $E_{molecule}$ are the energies of the SiO_2_ (100) surface substrate and isolated H_3_CCH(CH_3_)CHO, H_2_O, and O_2_ molecules, respectively. The energy directly calculated using VASP represents the zero-vibration energy. Considering a reaction temperature of 450.15 K, a vaspkit was used to correct the free energy of the molecule and adsorbed molecule [55,56,57].
(2)$$\rho_{ads}=\rho_{total}-\rho_{substrate}-\rho_{molecule}$$
where $\rho_{ads}$ is the total charge density of a molecule (H_3_CCH(CH_3_)CHO, H_2_O, or O_2_) adsorbed on the SiO_2_ (100) substrate, and $\rho_{substrate}$ and $\rho_{molecule}$ are the charge densities of the SiO_2_ (100) surface substrate and isolated H_3_CCH(CH_3_)CHO, H_2_O, and O_2_ molecules, respectively.

### 3.5. Chromatographic Mass Spectrometry Conditions

The chromatographic mass spectrometry conditions were established based on those previously reported with appropriate modification [58]. The carrier gas used was high-purity helium (99.999%) flowing at a rate of 1.8 mL/min. An injector temperature of 210 °C was employed. The initial program heating temperature was maintained for 5 min under 40 °C, with a heating rate of 30 °C/min; when the final temperature reaches 260 °C, it lasts for 5 min. The shunt ratio was 10:1. The ion source temperature was 230 °C, the transmission line temperature was 260 °C, and the ionization energy was 70 eV, using EI as the ionization mode. The ion monitoring mode was selected at 230 °C as the ion source temperature, and the solvent delay time was 20 s. Mass spectra were acquired within the range of 20–600 m/z at a scan rate of 500 amu/s.

## 4. Conclusions

In summary, we developed a CTL-based analysis method to measure trace IBL with a SiO_2_/MIL-53(Al) sensing platform. The chemiluminescence process was involved in the entire catalytic reaction and generated an excitation signal at 460 nm. This sensor system exhibits excellent selectivity, fast response, and a low operating cost. Importantly, between the CTL intensity and the low detection limit it was 0.49 ppm. When the sensor was put into the real sample analysis, the recoveries were in the range of 83.4% to 105%, with RSDs of 4.8% to 9.4%. The improved performance of the developed CTL can be attributed to several factors, including an enhanced specific surface area, improved electron mobility, and increased adsorption performance towards oxygen, as indicated by the results of the experiment and DFT analysis. We anticipate that this well-designed sensor will open up a promising scope for applications in other fields.

## Figures and Tables

**Figure 1 molecules-29-03287-f001:**
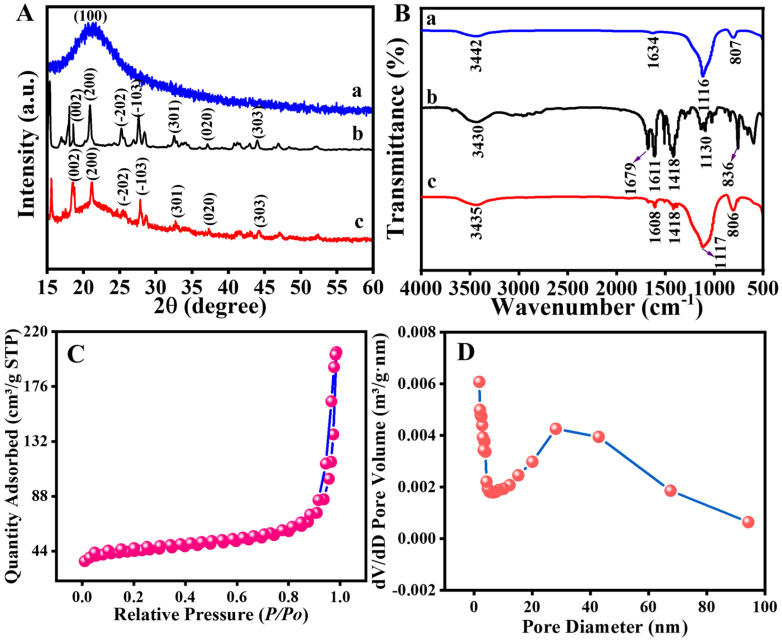
(**A**) Powder X–ray patterns of (**a**) SiO_2_, (**b**) MIL-53(Al), and (**c**) SiO_2_/MIL-53(Al). (**B**) Typical FT–IR spectra of (**a**) SiO_2_, (**b**) MIL-53(Al), and (**c**) SiO_2_/MIL-53(Al). (**C**) N_2_ adsorption–desorption isotherms of SiO_2_/MIL-53(Al) and (**D**) the pore size distribution of SiO_2_/MIL-53(Al).

**Figure 2 molecules-29-03287-f002:**
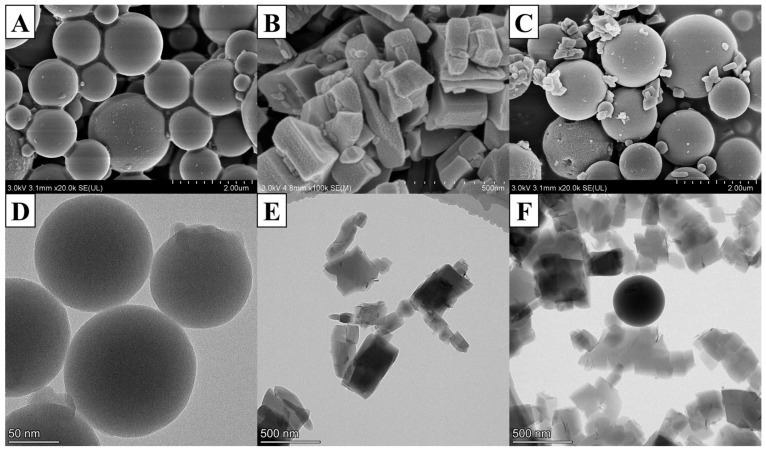
(**A**) SEM images of SiO_2_; (**B**) SEM images of MIL-53(Al); (**C**) SEM images of SiO_2_/MIL-53(Al); (**D**) TEM images of SiO_2_; (**E**) TEM images of MIL-53(Al); (**F**) TEM images of SiO_2_/MIL-53(Al).

**Figure 3 molecules-29-03287-f003:**
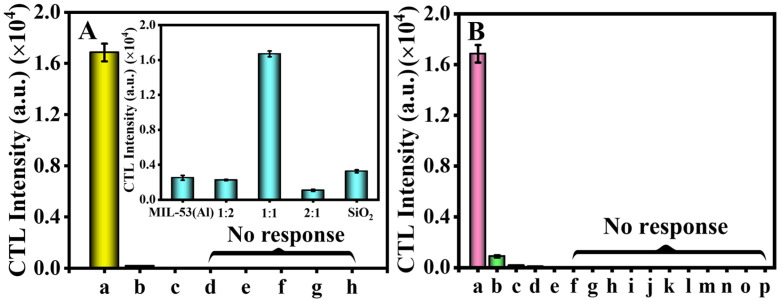
(**A**) CTL response of IBL on different catalysts surface: a, SiO_2_/MIL-53(Al); b, SiO_2_; c, MIL-53(Al); d, ZrO_2_; e, NiO; f, In_2_O_3_; g, Sn_2_O_3_; h, CuO, and (inset) proportion effect between SiO_2_ and MIL-53(Al); (**B**) CTL response of diverse gases on SiO_2_/MIL-53(Al): a, IBL; b, diethyl ether; c, ethyl acetate; d, propanol; e, acetone; f, ethylene glycol butyl ether; g, paraxylene; h, hexane; i, methanol; j, diethylamine; k, acetylacetone; l, n-pentane; m, methyl acetate; n, methanal; o, methylallyl aldehyde; p, glutaraldehyde. The concentration was 155 ppm, *n* = 7. Error bars represent ±SD (standard deviation). Experiment condition: The temperature is 177 °C, the wavelength is 460 nm, and the flow rate is 80 mL min^−1^.

**Figure 4 molecules-29-03287-f004:**
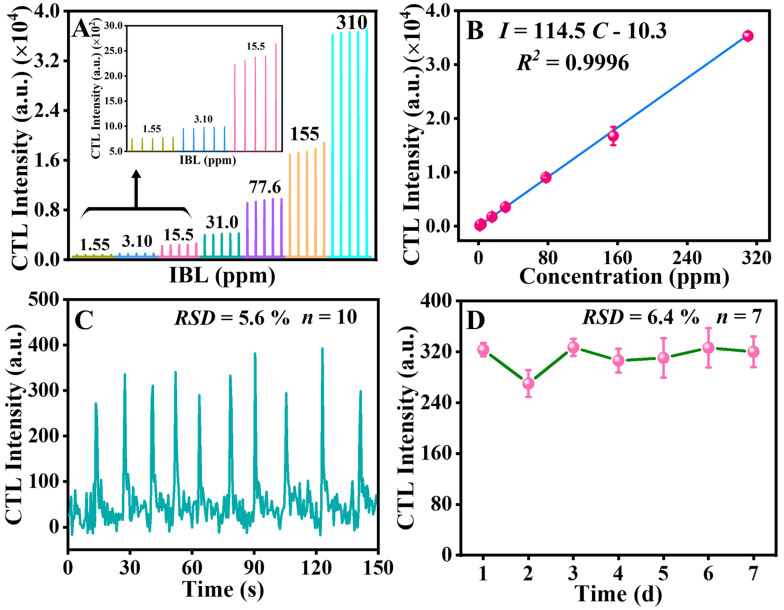
(**A**) CTL response curves for diverse concentrations of IBL on SiO_2_/MIL-53(Al). (**B**) Fitted equation between CTL emission intensity and IBL concentration. (**C**) The reproducible of CTL (IBL concentration: 3.10 ppm); (**D**) Stability of the CTL–based IBL sensor (IBL concentration: 3.10 ppm). Error bars represent ±SD (standard deviation).

**Figure 5 molecules-29-03287-f005:**
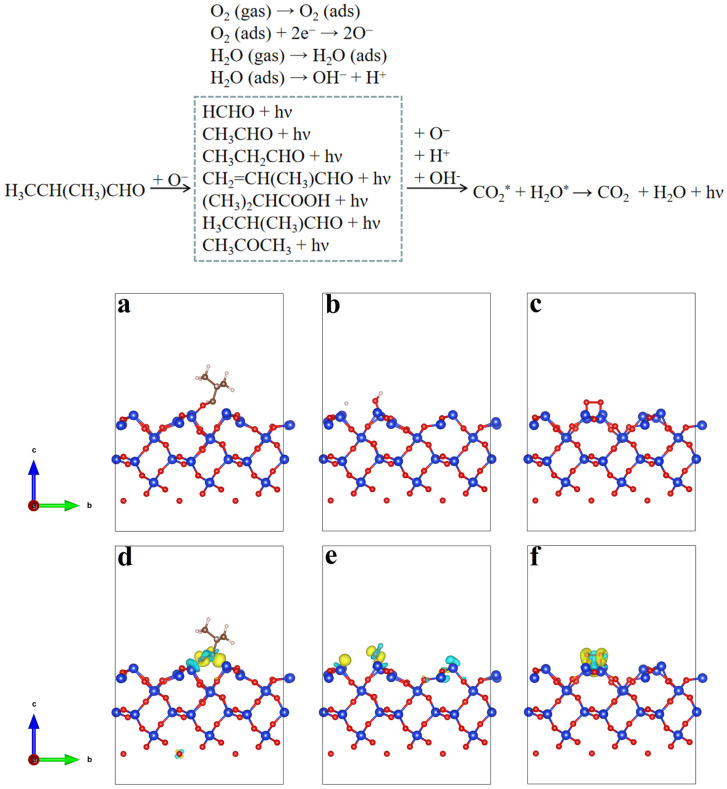
Images of (**a**) H_3_CCH(CH_3_)CHO, (**b**) H_2_O, and (**c**) O_2_ molecules after adsorption on the SiO_2_ (100) surface. (**d**) Charge density of H_3_CCH(CH_3_)CHO, (**e**) H_2_O, and (**f**) O_2_ molecules after adsorption on the SiO_2_ (100) surface. Isosurface value: 0.01 e/Bohr^3^; the blue area indicates a decrease in electron density, while the yellow area represents an increase in electron density.

**Table 1 molecules-29-03287-t001:** Analysis results of IBL samples (*n* = 7).

Sample ID	Mixture	Detected Values (ppm)	Spiked Concentration (ppm)	Detected Values (ppm)	Recovery (%)	RSD (%)
1	IBL	-	7.70	7.39	95.1	8.5
	Cyclohexanone		77.0			
	Butanal		77.0			
2	IBL	-	7.70	7.82	101	5.2
	Isobutanol		77.0			
	Ethyl acetate		77.0			
3	IBL	-	7.70	7.67	98.6	4.8
	N-propanal		77.0			
	Methylallyl aldehyde		77.0			
4	IBL	-	7.70	7.17	92.4	5.9
	Diethyl ether		77.0			
	Butanal		77.0			
	Methylallyl aldehyde		77.0			
5	IBL	-	7.70	7.29	93.8	6.4
	2-butoxy ethanol		77.0			
	Styrene		77.0			
	Methyl isobutyl ketone		77.0			
	3-methyl-2-butene-1-ol		77.0			
6	Volatile gas in a refrigerated display cabinet	4.25 ± 0.372	1.50	5.12	83.4	8.2
			3.10	7.26	97.3	7.3
			6.20	10.6	105	5.5
7	Volatile gas in medicine locker display cabinet	8.01 ± 0.248	4.00	11.4	92.2	9.4
			8.00	15.9	98.8	6.1
			16.0	24.1	100	5.7

## Data Availability

The data presented in this study are available on request from the corresponding author.

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
