# Peer review of "Enhanced Cataluminescence Sensor Based on SiO2/MIL-53(Al) for Detecting Isobutylaldehyde"

_molecules, 2024, doi:10.3390/molecules29143287_

Round 1

Reviewer 1 Report

Comments and Suggestions for Authors

The authors reported the detection of isobutyraldehyde using SiO2/MIL-53 via Cataluminescence. The submission is unacceptable for the following points:-

1.       The title should be revised to be informative reflecting the authors’ main findings. Remove redundant words such as ‘Highly Effective’; ‘Composites’.

2.       There is no sufficient explanation of why the composite of SiO2/MIL-53 is active compared to their components.

3.       Selectivity should be investigated using other interferences VOCs.

4.       Recyclability should be included.

5.       Real samples should be investigated.

6.       Experimental XRD and XRD pattern of PDF#85-1049 in Figure 1A.

7.       A comparison with previously published materials for IBL should be discussed and summarized in a Table.

8.       The language should be revised and typos should be corrected.

Minors

9.       Remove redundant words such as ‘novel’; ‘an essential’;

Comments on the Quality of English Language

Professional service for language editing should be included.

Author Response

Comment 1: The title should be revised to be informative reflecting the authors’ main findings. Remove redundant words such as ‘Highly Effective’; ‘Composites

Response: Thank you for the professional and valuable comments. We have revised our paper according to your suggestion.

Comment 2: There is no sufficient explanation of why the composite of SiO2/MIL-53 is active compared to their components.

Response: Thanks for the careful reading of our manuscript. In this work, compared with single MIL-53(Al) and SiO2, the advantages of using SiO2/MIL-53(Al) composites are its enhanced CTL performance for IBL, this part of the experiment can be seen in detail in Figure 3(A) of our experiment, which may be due to the synergistic effect between the different components of MIL-53(Al) and SiO2. [ X. Wang, T.K. Wang, G.K. Si, Y. Li, S.W. Zhang, X.L. Deng, X.J. Xu, Oxygen vacancy defects engineering on Ce-doped α-Fe2O3 gas sensor for reducing gases, Sensor. Actuat. B-Chem. 302 (2020) 1021–1031; Q.C. Zhang, Q. Zhou, Y.Wu, Y.X. Li, F.L. Tian, S. Tang, L. Jiang, Enhanced cataluminescence sensing of MIL-53(Al)/Sb2SnO5 composites for isobutanol detection, Meas. Sci. Technol. 34 (2022) 025106–025117].

Comment 3: electivity should be investigated using other interferences VOCs.

Response: We very much appreciate the careful reading of our manuscript and the valuable comments. We therefore conducted a supplementary experiment according to your suggestion. As shown in Figure S2, moreover, the CTL intensities of 2-butoxy ethanol and styrene were much weaker. As can be seen in Table 1, the experimental results showed that the other interferences VOCs do not cause a slight interference, including 2-butoxy ethanol, styrene, methyl isobutyl ketone, and 3-methyl-2-butene-1-ol. We have revised our paper accordingly.

Comment 4: Recyclability should be included.

Response: We very much appreciate the careful reading of our manuscript. Only one cataluminescence sensor based on SiO2/MIL-53(Al) has been used to complete this work, which could be reused more than 500 times with no significant changes in signal-to-noise ratio, and result indicated high stability.

Comment 5: Real samples should be investigated.

Response: We very much appreciate the careful reading of our manuscript. To assess the practical applicability of the developed sensor to IBL, two real gas samples, including volatile gas in a refrigerated display cabinet and medicine locker display cabinet were used to assess. The outcomes were delineated in Table 1, trace IBL in volatile gas in a refrigerated display cabinet and medicine locker display cabinet samples were determined in these samples were 4.25 ppm and 8.01 ppm, respectively. demonstrating recoveries spanning from 83.4% to 105%, accompanied by relative standard deviations (RSDs) ranging from 5.5% to 9.4%, which are deemed satisfactory.

Comment 6: Experimental XRD and XRD pattern of PDF#85-1049 in Figure 1A.

Response: Thanks for your comments on our paper. I apologize for not comprehending your intention, but we have nonetheless implemented additional enhancements and have revised our paper. (see Figure 1A in the marked manuscript).

Comment 7: A comparison with previously published materials for IBL should be discussed and summarized in a Table.

Response: The reviewer raises an interesting concern. It is a pity, to our knowledge, there are few materials for IBL in previously published work, only cataluminescence sensor based on Sm2O3 was used to determine IBL in References 7. [ Jiang L.; Wu Y.; Wang Y.; Zhou Q.; Zheng Y.G.; Chen Y.F.; Zhang Q.C. A highly sensitive and selective isobutyraldehyde sensor based on nanosized Sm2O3 particles. J Anal Methods Chem. 2020, 2020, 5205724].

Comment 8: The language should be revised and typos should be corrected.

Response: Thank you for the careful reading of our manuscript. We have carefully checked the manuscript for language problems and corrected them.

Comment 9: Remove redundant words such as ‘novel’; ‘an essential’.

Response: We very much appreciate the careful reading of our manuscript. We have removed the redundant words in our work according to your comments, such as “novel”

Reviewer 2 Report

Comments and Suggestions for Authors

The present manuscript deals with a cataluminescent sensor for detecting isobutylaldehyde. Based on a SiO2-MIL-53(Al) composite, the fabricated sensor presents a fast and reproducible response to the targeted VOC. Furthermore, the detection appears selective to isobutylaldehyde. Explanation of the sensing mechanism is attempted.

Despite a potential interest, the paper suffers from shortcoming and scientific inconsistencies. It is not suitable for publication. The statements are poorly supported by evidences. Some of my comments are:

The authors present SiO2 as a dopant for MIL-53(Al). In the present case, the sensor is a composite/mixture made of 2 different phases, which even do not appear entangled. It is not correct to speak about doping. Even more incorrect that the ratio is 1:1!

The FT-IR bands are not correctly assigned or the corresponding sentence is unclear. Where does N-H bond come from? The band at 3435 cm-1 seems to be assigned to different bonding in the same paragraph.

The authors claim that SiO2/MIL-53(Al) has the largest surface area and pore volume. These statements must be modified. MIL-53(Al) has higher SA and pore volume than the composite: to the best of my knowledge 304 m²/g > 138.9 m²/g!

The pdf card does not correspond to MIL-53(Al) but to Al(OH)3.

The observation/discussion of the microstructure must be revised. There is no difference of the SiO2 particle size and distribution in SiO2 and SiO2/MIL-53(Al) samples.

Why does SiO2/MIL-53(Al) display better catalytic effect of pure MIL-53(Al)?

The caption of figure 3 is incomplete.

What are the sensing conditions in figure 3b?

It would have been welcome to give a definition of cataluminescence.

Author Response

Reply to reviewer 2

Comment 1: The authors present SiO2 as a dopant for MIL-53(Al). In the present case, the sensor is a composite/mixture made of 2 different phases, which even do not appear entangled. It is not correct to speak about doping. Even more incorrect that the ratio is 1:1

Response: We very much appreciate for the careful reading of our manuscript. We do apologize due to ambiguous expression. SiO2/MIL-53(Al) was prepared via ultrasonication as follows: MIL-53(Al) (50.0 mg) and SiO2 (50.0 mg) were added to 4 mL of anhydrous ethanol and ultrasonicated for 30 min to achieve suspensions of MIL-53(Al) and SiO2, respectively. Subsequently, the SiO2 suspension was promptly added to the MIL-53(Al) dispersion and ultrasonicated for 30 min to achieve a uniform dispersion. The sensor material was then obtained through filtration and dried for 3 h under vacuum at 80 °C. Following this, 150 μL of ultrapure water was added, and the resulting products were thoroughly stirred to coat a cylindrical ceramic rod. Some CTL sensors with doping have a similar approach. [W. Ge, X. Zhang, X. Ge, K. Liu, Synthesis of α-Fe2O3/SiO2 nanocomposites for the enhancement of acetone sensing performance, Mater. Res. Bull.141 (2021) 111379–111388; Y. Hu, L. Li, L.C. Zhang, Y. Lv, Dielectric barrier discharge plasma-assisted fabrication of g-C3N4-Mn3O4 composite for high-performance cataluminescence H2S gas sensor, Sensor. Actuat. B-Chem. 239 (2017) 1177–1184; J.Z. Zheng, W.X. Zhang, J. Cao, X.H. Su, S.F. Li, S.R. Hu, S.X. Lia, Z.M. Rao, A novel and highly sensitive gaseous n-hexane sensor based on thermal desorption/cataluminescence, RSC Adv. 41 (2014) 21644–21649].

Comment 2: The FT-IR bands are not correctly assigned or the corresponding sentence is unclear. Where does N-H bond come from? The band at 3435 cm-1 seems to be assigned to different bonding in the same paragraph.

Response: We very much appreciate the careful reading of our manuscript. We do apologize for the error. We carefully checked it again and revised it in our manuscript. Related content is modified as follows: FT-IR spectroscopy was used to analyze the surface functional groups of the materials in Fig. 1B. The SiO2/MIL-53(Al) FT-IR spectrum demonstrated similar characteristic peaks to those of MIL-53(Al) and SiO2, including the presence of Si−OH groups adsorbed on the surface, a peak at 3435 cm-1 indicating the unique absorption of oh groups, −C=O asymmetric stretching at 1608 cm-1, −COO symmetric stretching vibration at 1418 cm-1, an asymmetric stretching vibration at 1117 cm-1 corresponding to oxygen void in bulk silicon, and a symmetric stretching vibration absorption peak of the Si−O−Si bond at 806 cm-1, respectively. [Kandilioti G.; Siokou A.; Papaefthimiou V.; Kennou S.; Gregoriou V.G. Molecular composition and orientation of interstitial versus surface silicon oxides for Si (111)/SiO2 and Si (100)/SiO2 interfaces using FT-IR and X-ray photoelectron spectroscopies. Appl. Spectrosc. 2003, 57, 628–635; Rahmani E.; Rahmani M. Al-based MIL-53 metal organic framework (MOF) as the new catalyst for Friedel-Crafts alkylation of benzene. Ind. Eng. Chem. Res. 2018, 57, 169–178].

Comment 3: The authors claim that SiO2/MIL-53(Al) has the largest surface area and pore volume. These statements must be modified. MIL-53(Al) has higher SA and pore volume than the composite: to the best of my knowledge 304 m²/g > 138.9 m²/g.

Response: We very much appreciate the careful reading of our manuscript. We have revised our paper according to your suggestion. The modification is as follows: The specific surface area (SBET), average diameter, and pore volume are shown in Table S1. The specific surface area of MIL-53(Al) is 304.03 m2/g, nevertheless, SiO2/MIL-53(Al) exhibited a higher BET surface area (138.92 m2/g) than that of SiO2 (3.81 m2/g), and the good surface area of SiO2/MIL-53(Al) provided an appropriate explanation for its excellent CTL activity in IBL-sensing.

Comment 4: The pdf card does not correspond to MIL-53(Al) but to Al(OH)3.

Response: We very much appreciate the careful reading of our manuscript. We do apologize for the error. We carefully checked it again and revised it in our manuscript. Related content is modified as follows: The crystal structures of SiO2, MIL-53(Al), and SiO2/MIL-53(Al) were examined by XRD. Both MIL-53(Al) and SiO2/MIL-53(Al) exhibit identical diffraction peaks at 18.03°, 23.50°, 25.21°, 27.41°, 33.80°, 37.28°, and 44.25°, consistent with those reported in [Ahadi N.; Askari S.; Fouladitajar A.; Akbari I. Facile synthesis of hierarchically structured MIL-53 (Al) with superior properties using an environmentally-friendly ultrasonic method for separating lead ions from aqueous solutions. Sci. Rep. 2022, 12, 2649–2665; Yan J.; Jiang S.; Ji S.; Shi D.; Cheng H. Metal-organic framework MIL-53 (Al): synthesis, catalytic performance for the Friedel-Crafts acylation, and reaction mechanism. Sci. China Chem. 2015, 58, 1544–1552]. The XRD patterns of SiO2 and MIL-53(Al) revealed their crystal structures (Fig. 1A (a, b)). Fig. 1A (c) illustrates that the XRD patterns of SiO2/MIL-53(Al) crystal planes, including (002), (200), (−202), (−103), (301), (020), and (303), show a resemblance to MIL-53(Al), suggesting that the crystal plane of SiO2/MIL-53(Al) remains largely unchanged despite the addition of SiO2.

Fig. 1. (A) Powder X-ray patterns of (a) SiO2, (b) MIL-53(Al), and (c) SiO2/MIL-53(Al).

Comment 5: The observation/discussion of the microstructure must be revised. There is no difference of the SiO2 particle size and distribution in SiO2 and SiO2/MIL-53(Al) samples.

Response: We very much appreciate the careful reading of our manuscript and proposed such valuable comments. We have checked and modified according to this opinion. The modification is as follows: It can be seen that the smaller MIL-53(Al) particles adhere to the larger SiO2 particles, which better fills the vacancy and makes the gas more in contact with the catalyst, thus improving its catalytic performance.

Comment 6: Why does SiO2/MIL-53(Al) display better catalytic effect of pure MIL-53(Al)?

Response: The reviewer raises an interesting concern and noteworthy concern. There are some similar literature reports [Y. Hu, L. Li, L.C. Zhang, Y. Lv, Dielectric barrier discharge plasma-assisted fabrication of g-C3N4-Mn3O4 composite for high-performance cataluminescence H2S gas sensor, Sensor. Actuat. B-Chem. 239 (2017) 1177–1184; N.M. Vuong, N.D. Chinh, B.T. Huy, Y.I. Lee, CuO-Decorated ZnO hierarchical nanostructures as efficient and established sensing materials for H2S gas sensors, Sci. Rep. 6 (2016) 26736–26748]. The possible reasons are as follows: 1). Synergistic effects: The interaction between SiO2 and MIL-53(Al) can lead to synergistic effects that enhance the overall catalytic activity of the material. 2). Improved stability: The presence of SiO2 can potentially improve the stability of MIL-53(Al) under reaction conditions, leading to better catalytic performance over time. These factors combined can contribute to the superior catalytic effect of SiO2/MIL-53(Al) compared to pure MIL-53(Al) or SiO2 alone.

Comment 7: The caption of figure 3 is incomplete

Response: We very much appreciate the careful reading of our manuscript. We have revised our paper.

Comment 8: What are the sensing conditions in figure 3b?

Response: We very much appreciate the careful reading of our manuscript. Figure 3b is performed under the sensing conditions of a temperature of 177 °C, a wavelength of 460 nm, and a flow rate of 80 mL/min.

Comment 9: It would have been welcome to give a definition of cataluminescence.

Response: We very much appreciate your valuable comments. According to your opinion, the revised description appears in “1. Introduction” in the revised manuscript.

Cataluminescence is a phenomenon where the emission of light is produced as a result of catalytic reactions occurring on the surface of a catalyst. This light emission provides valuable information about the nature and progress of catalytic reactions, making cataluminescence a useful tool for studying catalysis.

Round 2

Reviewer 1 Report

Comments and Suggestions for Authors

The authors addressed most of the comments and the revised version can be accepted. However, references Review for MOFs should be updated suggesting https://doi.org/10.1016/j.matchemphys.2023.127512; https://doi.org/10.1016/j.ccr.2021.214263; https://doi.org/10.1016/j.envres.2023.115349; https://doi.org/10.1016/j.ccr.2023.215043.

Comments on the Quality of English Language

Professional language should be conducted.

Author Response

Response to reviewer’s comments

Firstly, we are very grateful to editors and reviewers for the comments which would be valuable for the improvement of our paper. We have carefully considered the comments and have revised the manuscript accordingly. All the additions and modifications have been highlighted in red in the revised manuscript. Response to reviewer’s comments was explained point-to-point as following. Thank you for handling this paper again!

Reply to reviewer 1

Comments:The authors addressed most of the comments and the revised version can be accepted. However, references Review for MOFs should be updated suggesting. https://doi.org/10.1016/j.matchemphys.2023.127512; https://doi.org/10.1016/j.ccr.2021.214263; https://doi.org/10.1016/j.envres.2023.115349; https://doi.org/10.1016/j.ccr.2023.215043.

Response: Thank you for the suggestions. We have updated the references according to your suggestion.